# Conspiracy theories as engines of connection for enriched public debates on emerging technologies
Gabriel Dorthe [1,2] ✉

Conspiracy theories on COVID-19 mRNA vaccines and solar geoengineering (chemtrails) tend to reinforce one another, thereby posing significant challenges to public policy and scientific norms and generating confusion by conflating disparate issues. This paper is based on ongoing ethnographic fieldwork conducted in the United States, Germany, Switzerland, and France since 2015 in these two areas of active conspiracy attention, involving observation of social media pages and blogs, active participation in gatherings, and semi-structured interviews. Here, I adopt a diplomatic perspective, highlighting the reciprocal suspicion between science policy and conspiratorial thinking in a competition between two sets of connections of scientific facts, values, politics, fears, and hopes. The present study suggests that the contamination of the scientific discourse by seemingly unrelated claims in conspiracy theories offers fruitful insights to science communication into how publics make sense of science and technology in the fierce debates surrounding immunization and climate policy.

In June 2021, summer is just around the corner, and loosening restrictions are in the air. While the taxi driver shoves my bag into the trunk, I ask, somewhat ingenuously, if I must wear a mask in the car. "It really is up to you. I will wear mine, but I'm vaccinated." The COVID-19 pandemic has not only resulted in unprecedented restrictions on movement and social interaction but has also disrupted our routines and ways of communicating. Uncertainties and anxieties permeated the most mundane aspects of our lives, and fierce debates strained democratic societies. Who and what to trust? on what grounds? What can I do to protect myself? Charges and countercharges of deception are roaring in the public debate. We leave the train station, and after a few minutes of silence, stuck in traffic, he breaks the ice and asks what brings me to town. Idly gazing out the window, I falter: "an academic conference". Much more specific is his next question: "What do you think about vaccines?" This is a swift glide onto a treacherous terrain, no doubt. I lean towards the Plexiglas that stands between us.

What is my taxi driver expecting from me with his question? Why do I immediately feel uncomfortable, as if the conversation could further develop into an irreconcilable conflict of worldviews? And why do I feel pressured to have a clearcut opinion that I can express on the spot? The practice of ethnography has the effect of making my own academic positionality an intrinsic aspect of my study. Furthermore, my interlocutors frequently invite me to respond to my own research questions.

The most casual encounter can turn into a profound conversation about immunity and corporate power. Intimate circles are no safer than national elections, where public trust is condemned to be negotiated in a climate fraught with climate skepticism, vaccine hesitancy and conspiracy theories. Interdisciplinary scholarship[1–4], institutions and Internet platforms are engaged in a sustained effort to understand and respond to radical forms of suspicion against expertise-based public policy, widely despised as threats to public health, economic prosperity, and democracy[5–7].

In 1925, as the telegraph was entering the mainstream, the superintendent of the Eastern Division of the Associated Press warned about dishonest "fake news" that are misleading the public about important matters and spreading too fast for the traditional gatekeepers to assess the truth[8]. A century later, ubiquitous social networks and their sophisticated algorithms were matched with a strikingly similar tone when the director of the World Health Organization (WHO) declared: "We're not just fighting an epidemic; we're fighting an infodemic. Fake news spreads faster and more easily than this virus, and is just as dangerous"[9]. The reliability of information is a serious concern across the board, from public officials to those labeled "conspiracy theorists" who regularly tell me how difficult it is to identify honest substitutes to the "mainstream media", which they no longer trust. In addition to that, radical forms of public suspicion expressed in conspiracy theories meet a reciprocal suspicion from scientists and policy makers towards discordant claims.

This paper builds on ongoing fieldwork in two areas of active conspiracy attention: COVID-19 mRNA vaccines and the chemtrails narrative as a form of opposition to solar geoengineering (specifically stratospheric

[1]D-GESS (Department of Humanities, Social and Political Sciences), ETH Zürich, Zürich, Switzerland. [2]Research Institute for Sustainability at GFZ, Potsdam, Germany. ✉e-mail: gabriel.dorthe@gmail.com

https://doi.org/10.1038/s43247-025-02581-x                                                    **Article**

aerosol injection). Concerns towards these two apparently distinct (set of) technologies tend to mutually reinforce one another[10], as evidenced by a quantitative study of a large Twitter dataset[11]. Antivaccine activists often find out about projects aiming at controlling the weather while the chemtrail theory gets reinvigorated by controversies about vaccine safety and authoritative decision-making processes[12]. In what follows, I argue that the connections between topics and concerns are not merely random, nor trivial. Moreover, by establishing connections between issues that are unexpected by science communication, they bear affordances that portend further defiance of climate policies or public health decisions.

My research is based on ethnographic approaches and a set of methodologies that are particularly well-suited to understanding the problem, given their capacity to sit with ambivalence, identify unexpected connections (in contrast to survey work or direct interviews, which are predominantly used in the study of conspiracy theories). I have adopted "patchwork ethnography"[13], an approach and an attitude that acknowledges the growing constraints of academic mobility (lockdowns, travel restrictions, social distancing, shrinking research fundings) and transforms them into opportunities for a creative diversification of the sites of inquiry—from close ethnographic study of individuals and small communities to overview of the strategic institutional responses. I have conducted research online, in the United States, Germany, Switzerland, and France. Because they follow actors as they reflexively enact and react to the research problem under investigation, ethnographic approaches allow for the generation of new questions as the inquiry progresses, which may not be identified using purely theoretical or historical research methods.

Interpretatively, this paper contributes to the body of literature in the social sciences that examines how historical experiences and relationships between citizens and national and international institutions play a role in the formation, negotiation, and maintenance of trust. For example, vaccine hesitancy or conspiracy theories can thus be described as a reaction to prior political dynamics, often paralleling more sanctioned forms of activism[14–17]. The persuasive power of information is embedded in conflicting values, disenfranchisement and empowerment dynamics, not merely about what is true and what is not[18,19]. More specifically, I pay attention to the epistemic and the normative as being coproduced, engaged in a constant process of mutual stabilization[20]. In this perspective, rather than beliefs, as anthropologists have established[21], it is more insightful to study the collective practices of knowledge gathering, connection, and sense-making[22].

This paper offers a diplomatic perspective by contributing to a better understanding of the dynamics of suspicion in the public debates on highly contested emerging technologies. The philosopher of science Isabelle Stengers[23, VII] emphasizes the inherent discomfort of this position: speculating on an alternative outcome for a conflict always carries the risk of being seen as a traitor by both parties. While science communication and technology assessments tend to adopt a technology-specific approach, as exemplified by the WHO definition of vaccine hesitancy as "complex and context specific, varying across time, place and vaccines"[24, 4163], I argue that the contamination of the conversation by seemingly unrelated claims offer crucial insights into how publics make sense of science and technology. This may help institutions, science communication professionals, and involved scientists in navigating the muddy waters of public debates and, ultimately, in fostering public trust.

## Results and discussion
### Connections
Vaccine hesitancy has increased in alarming proportions since the COVID-19 pandemic and taken on a new set of dimensions. Long structured by concerns about aluminum or additives and autism[25], the advent of mRNA vaccines aggregated questions about the nature and impact of nanoparticles or graphene, and triggered conspiracy theories about the rationale behind their promotion as revolutionary. This form of radical skepticism has recently merged with conspiracy theories about climate modification. Since at least the early 2000s, activists rally under the portmanteau for chemical trails—chemtrails (as opposed to regular

condensation trails, contrails)—to expose and demand an immediate stop to stratospheric chemical spraying. They organize street protests, comment on online news articles, attend scientific conferences, and post countless videos and photos on social media. Despite the efforts of international organizations and scientists to debunk the claims[26,27], they maintain that solar geoengineering is already secretly deployed with adverse consequences for the health of humans, animals and plants. Until the pandemic reinvigorated it, the chemtrails theory was a well-established element of internet folklore[28], frequently employed as a rhetorical device for ridiculing irrational beliefs (akin to the notion of a flat Earth). The use of the word "chemtrails" is a point of contention within this community, some advocating for more scientific terminology, such as "geoengineering" or "climate engineering"[29,30]. Although this distinction is important, for the purposes of this article, the term "chemtrails" will be used, as the focus is on how conspiracy theories are regarded by the scientific and policy communities.

An interlocutor in the United States, for long heavily involved in the contestation of mRNA vaccines and global warming, was speaking for many when he told me that he only heard of chemtrails in the late Fall of 2023. On the other side, nanobots, 5G, or vaccines side-effects are extensively discussed in chemtrails groups. For instance, the French association *Ciel Voilé*, founded in 2011 to "study and expose the spraying of chemicals in the atmosphere"[31, my translation], switched gears in 2020–2021 and started to publish numerous articles and videos about threats to individual freedom, censorship and the dangers of vaccines. In 2023, the independent US presidential candidate Robert F. Kennedy Jr. (RFK Jr.), who has been described as an anti-vaccine activist[32], hosted Dane Wigington on his podcast[33], providing his website, *GeoEngineering Watch*, with an unprecedented political visibility. With the strapline "Geoengineering Affects You, Your Environment, and Your Loved Ones", this central resource for the opponents to solar geoengineering claims to have received over 50 million visits since its launch in 2009. As I will show, these two sets of so-called conspiracy theories share common features that allow cross-fertilization in the elaboration of worldviews that are primarily concerned with the (im)balances of power and see the individual body as both under threat and a reservoir for resistance.

The deepening asymmetry of power between citizens, and the State, multinational companies, international organizations, and billionaires is a core concern for those who see COVID vaccines as pushed through manipulative rhetoric. For example, the names of Anthony Fauci—Director of the US National Institute of Allergy and Infectious Diseases from 1984 to 2022—and Olivier Véran—French Minister of Social Affairs and Health from 2020 to 2022—, or entities such as Big Pharma or the World Economic Forum are often invoked[34–36]. In addition to being a vocal proponent of vaccines, Bill Gates is also personally involved in funding research on stratospheric aerosol injection[37,38], which makes him a credible villain for chemtrails activists. In Greece, the debt crisis that started in 2009 was instrumental in pushing people to adhere to the chemtrails narrative[39]. More broadly, psychological operations conveyed by the US army since at least the Cold War serve as a central reference for anti-vaccine activists to denounce propaganda and censorship[40,41]; and pesticides and agent orange are often mentioned by chemtrails activists[42,43]. The Philip Morris and Monsanto trials that made public how large companies have been circulating forged expertise to downplay the toxicity of their products[44,45] are cited by many of my interlocutors, who consider "why not again?" as a reasonable question to ask. These examples show that COVID mRNA vaccines and climate manipulation are connected to concerns about how key decisions for one's life and health are taken, and by whom.

### Curiosity
Set against opaque and abstract powers, conspiracy theorists' sense of truth is grounded in daily life experience. Chemtrails activists gladly present their engagement as originating in a disquieting observation of the sky, of their sky, above their home, take and share photos of aircraft trails as evidence of dubious human interventions[46]. Nine years later, a French interlocutor vividly remembers his first encounter with chemtrails, in June 2012, as a mind-boggling moment. Further research was needed to understand why

this overcrowded sky felt so upsetting to him. He encountered dozens of similar testimonies online, explaining that chemtrails last longer than contrails and are strangely shaped. They should thus be readily noticeable by everybody who would just "look up". Posters, stickers, hoodies, mugs, and buttons are available on e-commerce platforms such as Amazon or Etsy to help spread this watchword[47–49]. Hidden in plain sight, pervasive chemicals are traced back to their visual manifestations. For some, once properly trained, one can see nanoparticles used in the chemicals sprayed by airliners in how they react with sunlight[50]. Binding the visible and the invisible allows to make sense of concerns originating in emotional reactions while contemplating the sky.

If pictures of skies are the most common type of evidence produced by chemtrails activists, they also collect and analyze rainwater and soil samples from their gardens to document the presence of heavy metals sprayed above their home[51,52]. In his influential book on geoengineering, Oliver Morton notes that chemtrails activists are genuinely worried about dangers posed by climate manipulation for their health[53, 102–104]. One of the first things that the aforementioned newcomer to this community told me was that he would like to contribute to Wigington's website GeoEngineering Watch with data from his location. Such practices echo citizen science in toxic environments: "Environmental justice activists typically adopt dual orientations towards science, of mistrust and reliance: (1) challenging the methods, questions, and uses of science, particularly in the context of vested corporate interests, while (2) relying on science itself, as a necessary tool to make investigations, provide evidence, and make arguments"[54, 10]. While experimental sciences struggle to provide a detailed picture of the pervasiveness of toxic chemicals and heavy metals in the environment[55], they rely on activist movements to raise local pollution issues[56]. Instead of a clear distinction between legitimate environmental activists and irrational conspiracy theorists, there is a permeable circulation between science and embodied experience, as exemplified by the public discussion between Wigington and RFK Jr., who initially gained recognition as an environmental lawyer[57].

Feeling directly threatened by air pollution and toxic materials, chemtrails and vaccine conspiracy theorists view their bodies and minds as a front line in the battle against malicious powers. Stories of people suffering adverse effects (often referred to as "vaccine injuries") from vaccination abound: athletes experiencing a stroke during a game or relatives who have developed allergies, heart attacks, or other health complications. In the chemtrails community, many attest to experiencing adverse health effects from the unjustly sprayed chemicals, in testimonies[58] that bear resemblance to cases of chronic fatigue syndrome and multiple chemical sensitivity, which sufferers "have to fight to get" due to the lack of recognition by medicine and the exclusion from insurance categories[59].

These activists thus evince a pronounced curiosity about the techniques and products that can preserve and enhance their physical and mental health, such as detoxification, meditation practices, and organic food. Some experiment with crystals, such as orgonite or shungite, used by people concerned about electromagnetic waves. The COVID-19 pandemic has highlighted the importance of understanding what can either boost or impair immunity. Many individuals thus started experimenting with a range of dietary supplements (e.g., zinc, vitamins, probiotics), homeopathic medicine, or other alternative therapies. The National Institutes of Health offers a comprehensive fact sheet on this subject[60], which explicitly states that they can only provide a supportive environment for vaccines. Skeptics posit that natural immunity can rather function independently, as evidenced by the widespread dissemination of an undercover video of a Pfizer scientist admitting to this fact[61], triggering months of debates, from conspiracy-inclined groups to parliaments.

These techniques are seen as a means of reasserting the capacity of individuals to take care of themselves, to reframe what counts as immunity in a specific context that makes sense to them, while public health policies and toxicity studies tend to focus on specific aspects cut off from the complexity of people's lives. Organized in response to the World Economic Forum's "Great Reset" framework introduced during the pandemic[62], "The Greater Reset" conference series is structured around the premise that the

problem—the power grab of global elites over citizens through technocratic control—does not require further diagnosis but rather calls for the implementation of concrete solutions. However, in many instances, heightened alerts are prioritized over the identification of solutions, illustrating a tension between the global nature of the issue and the capacity of individuals or groups to empower themselves. The Facebook group Chemtrail Cures was created in 2017 to exchange "on what helps humans and your animals fight off the illnesses that Chemtrails are bringing to world, to your neighbourhood, to your home"[63]. Its members mostly share photos of skies and other information meant to raise awareness about chemtrails and weather manipulation, just like on any similar social media page.

The idea of being sprayed predates contemporary scientific conversations about solar geoengineering and was initially associated with military projects in the Vietnam War, where the US Air Force used cloud seeding techniques as weapons[64]. As secretive military agencies gave way to elite academic institutions, chemtrails activists began to pay close attention to the scientific literature. The widely cited documentary What in the sky are they spraying? features some of them at the 2010 Meeting of the American Association for the Advancement of Science, where they posed questions to prominent figures in the field of geoengineering[65]. In 2014, two activists from the United Kingdom attended the Climate Engineering Conference 2014 in Berlin, surprised to be accepted by the organizers, happy that the event helped them understand the science better, and proud to be part of the broader conversation[66]. Those who oppose vaccination employ similar efforts to demonstrate the potential dangers of nanoparticles in mRNA vaccines: they circulate and discuss scientific papers, preprints, graphs, and microscope renderings that bear the weight of evidence and are expected to bring credibility to their broader concerns. Talking to conspiracy theorists can be overwhelming because of the amount of scientific evidence that they provide. My interlocutors often look surprised when I confess that I have not heard of this study or that report, as if I was missing the point entirely. In other words, these skeptics are far from being anti-science, even if they may, at times, misunderstand or mischaracterize the information provided in scientific papers. They invoke the authority of science and facts to contest authorities that are perceived as illegitimate or as having abused their power.

## Boundaries

The alarm raised by the WHO director about the "infodemic" demands a forceful response: "Now more than ever is the time for us to let science and evidence lead policy. If we don't, we are headed down a dark path that leads nowhere but division and disharmony."[9]. Vibrant calls from public officials and scientific institutions for the preservation of the authority of science and expertise are nothing but new. They often connect this authority to core values that science is expected to foster, such as social progress, democracy and economic prosperity, as exemplified by a 2001 Nature Medicine editorial that urged the scientific community to "remain vigilant in order to preserve the trust and value that society affords scientific progress."[67]. The erosion of trust thus directly threatens the benefits of science. Such appeals therefore carry with them an underlying fear of irrational reactions from certain segments of the public and draw boundaries between acceptable and detrimental objections.

In the early 2000s, public trust in science was indeed increasingly tested by the advent of a regime of technoscientific promises[68–71], welding research to values and imaginaries of a desirable future enabled by progress in science and technology[72]. In the context of limited funding resources and a concomitant rise in research costs, promises are performative acts facilitating the securing of competitive funding, media attention, and public support. While grand narratives of power and prosperity tend to present the future as univocal[73,74], they can turn into threats for some parts of the public who feel disempowered or dispossessed from their aspirations[69, 292].

Marking a milestone in the development of nanotechnology promises, the report published in 2002 under the auspices of the National Science Foundation and the Department of Commerce mapped the convergence of nano, bio, info and cognitive (NBIC) technologies with the objective of "Improving Human Performance" through the use of new materials,

biomedicine, brain-computer interfaces and nanobots[75]. The latter, minuscule devices acting like computers that could be injected to deliver drugs with perfect precision, cure diseases and ultimately make humans immortal, have been promised by prominent scientists and engineers since the 1980s[76]. They were still regarded as "futuristic outcomes" in 2002[75,410] and remain so today, despite enduring promises as evidenced by publications on nanobots as "an alternative for treatment to vaccines for COVID-19"[77].

During the COVID-19 pandemic, conspiracy theorists struggled to make sense of the nanotechnology incorporated into mRNA vaccines. They easily identified engineers, futurists, and publications that have been extensively speculating about the technological transformation of humanity[78,79]. The NBIC report was already concerned that the uncritical celebration of nanobots might undermine the credibility of nanotechnology as a whole, "feeding further the fears that the products of our creation may be smarter than we are and that we may sow the seeds of our own destruction"[75,376]. The concern that public misunderstandings might lead to rejection responses is indeed frequently identified as a threat to the development of promising technologies[80]. For a nanotechnology pioneer, "people concerned about technology and the future are a limited resource. The world cannot afford to have their efforts squandered in futile campaigns to sweep back the global tide of technology with the narrow broom of Western protest movements. The coming problems demand more subtle strategies."[76,167].

The climate crisis gives rise to comparable demands for urgent action and a contained public debate. In light of the inadequacy of intergovernmental negotiations to reduce greenhouse gas emissions, a set of radical measures are gaining traction within the scientific community[81,82]: geoengineering, defined as the "deliberate large-scale manipulation of the planetary environment to counteract anthropogenic climate change"[83]. Arguably the most controversial of the proposed solutions, solar geoengineering involves the spraying of particles into the stratosphere, forming a thin layer of aerosols capable of reflecting a portion of the sun's radiation. The Nobel Prize laureate Paul Crutzen is known for "breaking the taboo" on this idea[84] in his *Nature* opinion piece that popularized the concept of the Anthropocene, the proposed new geological epoch in which humanity is the primary geological force. He concluded with a stark warning: "A daunting task lies ahead for scientists and engineers to guide society towards environmentally sustainable management"[85]. In a more detailed account, he further posited solar geoengineering as a way to "resolve a policy dilemma" between the reduction of air pollution (and its albedo effect) and the control of global warming[86].

Notwithstanding the profusion of research on the ethics and governance of solar geoengineering, the scientific literature on the subject continues to be pervaded by skepticism regarding the capacity of democratic societies to make prompt decisions on urgent matters. Reactions from the public that could impede research on this contentious topic are of primary concern. In April 2024, a team from the University of Washington conducted the first outdoor test in the United States of a cloud brightening technology. The *New York Times* article that revealed the experiment essentially framed it as a conspiracy, stating that the research team had "kept the details tightly held, concerned that critics would try to stop them."[87].

While solar geoengineering researchers are accustomed to addressing impassioned public responses to their work, the chemtrails community presents a distinctive challenge. David Keith notes that he has received many extreme critiques and "two death threats that warranted calls to the police" from "people who are convinced by the chemtrail conspiracy theory"[88,125]. This segment of the public gives rise to an even more existential suspicion among scientists, for whom such bloated controversies risk suppressing research and the patient assessment of benefits and risks[88,17]. For others, they provide superfluous confusion in an already impassioned debate[28] or distraction from real problems[27]. In sum, conspiracy theories are viewed as a threat to effective governance, because they "have the ability to distort otherwise rational conversations and debates about complicated topics such as solar geoengineering"[89,132]. The differentiation between rational and irrational oppositions allows experts in the field to define the terms of the public debate in a manner that aligns with their expertise and perspective. In this vein, Keith advocates a disaggregation of expert claims to clarify the

conversation and avoid polarizing positions[90]. However, this may result in the premature disqualification of concerns from segments of the public pertaining to the impact of technoscientific schemes on the quality of the air they breathe, the food they consume, the climate they inhabit, or the information they rely upon.

## Conclusions

Emerging technologies trigger suspicion in the public (which can manifest in various forms, including conspiracy theories), as well as among experts who strive to maintain control over the terms of the debate. The profoundly relational and intricate phenomenon of trust and distrust[91,92] is not only a question for publics, but also for scientists and institutions. This article argues that concerned publics establish connections between issues that may be unexpected from science communication, debunking or prebunking[93,94] perspectives. My taxi driver, who was hesitant about getting vaccinated against COVID-19, would have been an ideal candidate for these approaches. However, they proved ineffective, as the connections he relied on to make his decision were not the same as the ones between a dangerous virus and an inoculation technique. For him, trusting science had very little to do with the technicalities of mRNA or complying with experts' advice. But a great deal with how his personal experience could relate to the people behind the needle.

I begin to cautiously explain that my job is precisely to understand the debates on the COVID-19 vaccines… The taxi driver interjects with ease: "I've already told you that I'm vaccinated, right? But you know what? I used to be opposed to vaccines." Why did he change his mind, in the early days of vaccines rollout? He went online to learn more about these new vaccines. He found out about this husband-and-wife team of scientists who are fully devoted to science, as he goes on while the traffic gets more fluid, reinvesting all their money in research, to the point of living in the modest flat they shared since they were students. Having no idea if this is true or not, I keep listening to him. "They are German, you know. I mean, originally from Turkey, but they are German". The precision is meaningful to him, being an Italian immigrant to France, which I could not have guessed given his absence of an identifiable accent. Knowing who and what he was dealing with, he went for the Pfizer-BioNTech vaccine.

Responses to conspiracy theories usually take the terms of the topic for granted instead of embracing the potential for unexpected and creative connections. If scientists and policy makers wish to enrich their understanding of marginalized citizens, rather than seeing them as ignorant and irrational[95,894], listening to how contentious issues matter to people is crucial. Yet this is not enough. Neither technoscientific promises nor conspiracy theories speak in a general sense. Rather, they engage a specific understanding of the world in which some issues are deemed more pressing than others.

Nanotechnology and solar geoengineering are framed as far-reaching responses to global challenges (economic growth, global health, climate policies). Conspiracy theorists see them as threats to their bodily autonomy and individual freedom. These two sets of connections are either too wide or too narrow to enter a productive dialog. They both need to be entangled into creative frameworks of robust meaning-making processes, where productive perspectives about science, individual bodies, and society are celebrated and encouraged. This article is a plea for a shift in focus from particular issues to the reticular dynamics of connections between them. As the first scholarly account on chemtrails already indicated[29,81], this may not pacify the debate on current issues, but it will undoubtedly enhance the climate in which science and technology address intimate and global concerns.

## Methods

This study employs a qualitative mixed methods approach that intersects four distinct types of materials:

(1) Observation of social media pages and blogs, 16 for chemtrails and solar geoengineering conspiracy theories (since 2015), 15 for COVID vaccines (since 2021). See full list in Supplementary Materials. The selection of pages was conducted using a snowball technique, with an effort to balance the centrality of key US-based accounts with geographical diversity. Extended observation periods facilitate the

discernment of the stability or shifts in focus of the studied communities, particularly the escalating interest in vaccines within the chemtrails community, which serves as the focal point of this article. I record and store posts when they are particularly illustrative of the ongoing conversation.

(2) Monitoring of extant literature on conspiracy theories, along with strategies for combating or debunking them. With this material, I aim at gaining a comprehensive understanding of the broader structure of the debate on the topic, and secondly, to identify the extent to which my approach diverges or aligns with the prevailing framing. To this end, I professionally follow the literature in STS, and focus on two primary sources that offer insight into the dynamics of the conversation in policy circles: the *WHO Infodemic Management newsletter*, published on a biweekly basis, and the *Harvard Kennedy School Misinformation Review*.

(3) Participant observation of online and in-person gatherings. In January 2023, I attended the fourth edition of "The Greater Reset" conference series in Bastrop, Texas, United States. This 5-day in-person event provided a valuable opportunity to refine my approach through active engagement with the 300 participants, including one-to-one or small groups discussions and listening to the talks. I did not conceal my positionality as a postdoctoral researcher affiliated with the Program on Science, Technology and Society at Harvard University and the Research Institute for Sustainability—Helmholtz Centre Potsdam; rather, I carefully disclosed it when the conversation allowed to understand better how my interlocutors relate to elite academic institutions and how they view my participation in an event of this nature. A detailed account of this experience is the topic of another paper currently in development.

(4) Semi-structured interviews are conducted with individuals to explore specific aspects in greater depth. 11 individuals were identified through the snowballing technique, a method that proved crucial to gain trust and encourage individuals to engage in conversation. In some cases, the conversation was reconducted several times to allow my interlocutors the freedom to address and elaborate on what they consider important, so that I can gain insight into how they connect various topics and learn how they make sense of information. When feasible, the conversation has been audio-recorded, and a consent form signed by the interviewee. However, given the nature of the inquiry, it is far from trivial to note that this procedure is often not applicable. Insightful conversations can occur in unexpected places or moments: on the street, at the hairdresser's, or in a taxi—as the opening vignette of this article illustrates. In such cases, as in the present article, I utilize indirect voice or recall the content of the conversation without aiming at exact citation. The quotes in the vignette are reconstructed from my notes.

After initial review, and under strict guarantees of confidentiality regarding the interviews, the Harvard University IRB exempted this study on June 6, 2022 (Protocol # IRB22-0301).

## Data availability

The content of the semi-structured interviews is confidential. Otherwise, no new data were generated.

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

## Acknowledgements

This research has been supported by the Harvard Data Science Initiative, Project on Trust in Science, the Research Institute for Sustainability at GFZ Potsdam, and the Swiss National Science Foundation (P2LAP1_187760). I am deeply grateful to Mark Lawrence for his support over the years and for encouraging me to write this paper. For the past decade, Sheila Jasanoff has provided unwavering mentorship and intellectual sparring, offering vital support for my (occasionally offbeat) explorations. Stefan Schäfer and Sam Weiss Evans played a crucial role in helping me to think more deeply and carefully about this paper. The elaboration of this paper has further benefitted from joyful and insightful conversations with Gretchen Bakke, Bernadette Bensaude-Vincent, Margarita Boenig-Liptsin, Damien Bright, Marco dell'Oca, Martín Fonck, Cameron Hu, Janel Jett, Anna Lytvynova, Mariam Mauzi, Fabienne Ruppen, Melissa Salm, Hilton Simmet, Vidya Subramanian, and the 2021–2022 cohort of the Harvard STS Fellows. I am also grateful to three anonymous reviewers for their useful comments on earlier versions of this paper, and to the journal's editor, Heike Langenberg, for her invaluable help in navigating the editorial process.

## Competing interests

The author declares no competing interests.
