## [Transparent Peer Review file · Communications Earth & Environment]

Conspiracy theories as engines of connection for enriched public debates on emerging technologies

Corresponding Author: Dr Gabriel Dorthe

Version 0:

Decision Letter:

Dear Dr Dorthe,

Your manuscript titled "Aerosolization of trust in science" has now been seen by three reviewers, whose comments are appended below. You will see that they find your work of some potential interest. However, they have raised quite substantial concerns that must be addressed. In light of these comments, we cannot accept the manuscript for publication, but would be interested in considering a revised version that fully addresses these serious concerns.

We hope you will find the reviewers' comments useful as you decide how to proceed. In particular, we would like to endorse reviewer 2's suggestion: please consider carefully whether you would like to revise the manuscript (1) as a Perspective (a short, somewhat opinionated literature review), or (2) as an original research article.

If you opt for (1), please clarify the new conceptual framework that arises from the literature synthesis, ensure it is fully supported, and deepen your discussion of the implications and recommendations that follow from your analysis, beyond the notion of dialogue between natural and social scientists and between scientists and the public. In this case we would recommend removing reference to fieldwork.

If you opt for (2), please provide full methodological details of your fieldwork and discuss the results in more detail, to clarify how your suggested framework of thinking is supported by the empirical work.

Should additional work along these lines allow you to address all the reviewers' criticisms, we would be happy to look at a substantially revised manuscript.

If you choose to take up this option, please either highlight all changes in the manuscript text file, or provide a list of the changes to the manuscript with your responses to the reviewers.

If the revision process takes significantly longer than three months, we will be happy to reconsider your paper at a later date, as long as nothing similar has been accepted for publication at Communications Earth & Environment or published elsewhere in the meantime.

We understand that due to the current global situation, the time required for revision may be longer than usual. We would appreciate it if you could keep us informed about an estimated timescale for resubmission, to facilitate our planning. Of course, if you are unable to estimate, we are happy to accommodate necessary extensions nevertheless.

Please use the following link to submit your revised manuscript, point-by-point response to the reviewers' comments with a

list of your changes to the manuscript text (which should be in a separate document to any cover letter) and any completed checklist:

Link Redacted

Please do not hesitate to contact me if you have any questions or would like to discuss the required revisions further. Thank you for the opportunity to review your work.

Best regards,

Heike Langenberg, PhD
Chief Editor
Communications Earth & Environment

On Twitter: @CommsEarth

EDITORIAL POLICIES AND FORMAT

If you decide to resubmit your paper, please ensure that your manuscript complies with our editorial policies and complete and upload the checklist below as a Related Manuscript file type with the revised article:

Editorial Policy Policy requirements
(Download the link to your computer as a PDF.)

For your information, you can find some guidance regarding format requirements summarized on the following checklist: (<https://www.nature.com/documents/commsj-phys-style-formatting-checklist-article.pdf>) and formatting guide (<https://www.nature.com/documents/commsj-phys-style-formatting-guide-accept.pdf>).

REVIEWER COMMENTS:

Reviewer #1 (Remarks to the Author):

This perspective piece argues that conspiracy theories need to be studied not just as manifestations with concerns about “trust in science”, but that they have to do with norms and values about the power of individual bodies; that they can be seen as “manifestations of disenfranchisement towards scientific promises.”

I am not sure, frankly, how novel the main claim is, as I have not read everything in the conspiracy theory literature within the field of communications / media studies. There are a few different fields which may have treated this topic (sociology, anthropology, comms, STS, and so on). Certainly, within science and technology studies (a field I am more versed in), there are discussions not just about trust in experts, but in power relations. So the more general argument in the paper may have been made.

However, it is new to study these two examples – solar geoengineering / chemtrails and antivaccines – together. In my view, this makes it enough of a novel contribution to be worth publishing, inasmuch as it could help scientists (who seem to be an intended audience) working in these fields understand some of the responses to their work. (I would note that phrases like “entanglement of norms and values with expertise” might be a bit intangible for this audience, and that it would be worth scanning the paper for social science jargon.)

My main, easy-to-address comment is that it would be helpful to have a bit more about the methods / fieldwork up front – even though it’s a Perspective, it alludes to empirical material, and that would be useful to contextualize (when, where, etc.).

A second concern with the manuscript as it is is that it takes some time to describe solar geoengineering and make reference to that, but actually, I believe the chemtrails conspiracy predates discussions of solar geoengineering and arose independently of it. Then, the chemtrails community latched onto solar geoengineering as it emerged. This could be an important point for the argument (and a way that it is different from the mRNA nanobots) – I don’t think the chemtrails derived from the technoscientific promises of geoengineering. People thought they were being sprayed for a variety of purposes, mind control, and so on – solar geoengineering entered much later as an additional rationale for the spraying. I am not sure that anyone has written an authoritative history of chemtrails, though, so I can’t back this up with a citation – this is simply my understanding from following the topics for many years.

A third comment is that I found myself wishing for a stronger conclusion that could advise the reader about how “a comprehensive dialogue between social and natural sciences, and between scientific communities and the broader public”, that cultivates robustness of discourses, etc., could happen – what are the specific forms or policy actions or cultural

changes needed, who does what to develop this, etc.

In summary, I think this essay could be a useful contribution with some modest changes. The main point – that it's not just about "trusting the science" - is one that needs to be made in a variety of ways for people to truly operationalize it, and so I think it is worth making it for this audience, with regards to these two topics.

Minor points:

- The sentence on p. 9, line 269 seems incomplete or jarring; consider rewording, and I assume the word "Respond" should be "Responses."
- p. 10, line 297-8 reminds me of Jill Lepore's book *These Truths*, which is worth reading.
- The title and the concluding sentence with the metaphor of aerosolization just don't work for me; I wish they did and I can't explain why they don't.

Something of an aside for the authors to consider: The point on p. 6 about the sharp opposition between science and anti-science being misleading, and about conspiracy scientists being interested in science, is important. I am reminded of the observation by Thom Davies and Alice Mah in their book *Toxic Truths: Environmental Justice and Citizen Science in a Post-Truth Age*: "Environmental justice activists typically adopt dual orientations towards science, of mistrust and reliance: (1) challenging the methods, questions, and uses of science, particularly in the context of vested corporate interests, while (2) relying on science itself, as a necessary tool to make investigations, provide evidence, and make arguments" (2020: 10). I suppose the authors might rather separate this population of study – chemtrailers and anti-vaxxers – from EJ activists and citizen scientists, but the ways in which science and "lived experience" are in tension in both areas are kind of interesting, and why some people become or are seen as activists and citizen scientists rather than conspiracy thinkers would be an interesting area of study. There is a lot to say about the intersection of both these with alternative medicine, too, which p. 8 touches on a bit, but there's perhaps more literature there.

Reviewer #2 (Remarks to the Author):

Thank you for the opportunity to review, "Aerosolization of trust in science." On the one hand, the manuscript has the potential to make an important contribution to the literature on the social dimensions of conspiratorial thinking, public perceptions of science, and science communication. However, I don't think the "perspective" format is an appropriate model to do this effectively for reasons explained below.

The manuscript does not engage deeply with literature on the social aspects of geoengineering (GE), conspiracies about chemtrails, vaccine-related conspiracy theories, or conspiracy theories in general. This is somewhat understandable given that it is a "perspective" rather than an article. However, the brevity of engagement with existing literature is coupled with a lack of evidence about the central argument of the paper, that:

"Most researchers and university administrators are familiar with that phenomenon, but, while overdetermining the future with grand narratives of power, promises might instill a feeling of dispossession in some parts of the public, who might struggle to make sense of them, or defy them as yet another ideological instrument for maintaining power imbalances (Adam and Groves 141 2007; Santos 2014). Crafted to generate trust and faith in progress, they turn into threats due to the univocity of this progress: "Elevated levels of expectation and confidence also have the effect of inflaming concerns about risk in different communities based on differing values, knowledges or institutional and organizational forms" (Borup et al. 2006, 292). Conspiracy theories about nanobots in mRNA Covid-19 vaccines, seen in this light, derive from the promises made by scientists and entrepreneurs, imbuing fears that advanced technologies might be implanted in bodies without consent."

My largest concern with this paper is there isn't enough evidence presented that would convince readers that this is the dynamic behind conspiracies surrounding chemtrails and/or vaccines. What is the basis for arguing that overhyping nanotech or solar geoengineering is the cause of conspiracies about these and related technologies?

For example, the following argument is made without any evidence, in the form of citations to previous studies or to primary data collection: "They also play a close attention to the vast literature that is enthusiastically promising nanorobots or praising the merits of graphene as a revolutionary material. A sharp opposition between science and anti-science is thus misleading, as a close look at the communities accused of conspiracy theories shows that they nourish a keen curiosity towards scientific announcements and publications, sometimes at the expense of taking them at face value—while they might be preliminary, falsifiable, or sometimes fraudulent."

The softer claim seems more defensible: "In what follows, I explore how differing values or visions of collective progress shape the reception of technoscientific promises, and how conspiracy theories reframe them in terms of health hazards and political disempowerment." Even if this line of argument is more defensible, there still isn't enough evidence provided. For example, there is evidence provided, via references to past research, that perceptions of real power imbalances fuel conspiratorial thinking.

This would be a great perspective piece if it took the form of either a (1) short literature review as illustrations for a new conceptual framework or (2) a short but highly empirical piece for arguments to emerge from data. Instead, it sits in between, failing to establish empirical credibility or conceptual richness. This is not a criticism of the argument itself, which seems novel and important. The issue is that the perspective format is not conducive to making this argument.

Other comments:

This sentence from the abstract is confusing: "Building on field work in two communities that tend to mutually reinforce each other, solar geoengineering opponents known as chemtrails and anti mRNA Covid vaccines,"

p. 2 – "the same technoscientific promises can be seen as threatening and enslaving and tend to form the bedrock of conspiracy theories." I would change to "some" conspiracy theories. I can think of many conspiracy theories that are not responses to technoscientific promises.

p. 4 "have been promised since the 1980 by prominent scientists and engineers" – 1980s

Reviewer #3 (Remarks to the Author):

The paper builds on two cases of conspiracy theory – one related to solar geoengineering and another to Covid vaccines – to offer lessons for scientists on how to navigate social discourse and gain public trust. It certainly is an important issue worth exploring and discussing. Overall, the author stresses the importance of a dialogue between social natural sciences and suggest discourse cultivation on all sides.

The paper makes some interesting points, but readers would be better served by reading Cash et. al (2003) or Cash and Belloy (2020). This paper unfortunately does not offer enough of new thoughts/insights.

Cash, David and Clark, William C. and Alcock, Frank and Dickson, Nancy M. and Eckley, Noelle and Jäger, Jill, Saliency, Credibility, Legitimacy and Boundaries: Linking Research, Assessment and Decision Making (November 2002). Available at SSRN: <https://ssrn.com/abstract=372280> or <http://dx.doi.org/10.2139/ssrn.372280>

Cash, D.W., & Belloy, P. (2020). Saliency, Credibility and Legitimacy in a Rapidly Shifting World of Knowledge and Action. Sustainability.

Comments

Line 11: instead of apprehended I would use comprehended/understood.

Line 33: anti-mask (delete "s")

Line 41: references?

Line 46: It reads as if you equate "chemtrails" and "solar geoengineering".

Line 49: chemical safety (delete s)

Line 52: comprehend instead of apprehend.

Line 61: delete: "social sciences (a field named as"

Line 65: delete "whom", delete "trustful"

Line 78: what text?

Line 82: "on science" instead of "in"

Line 97-98: grammatical error.

Line 111 to 112: transition needed.

Line 112: delete "Still"

Line 123: "shattering the credibility of nanotechnology" I don't this is a result

Line 131: "the" instead of "this"

Line 138: "overdetermining the future" – what do you mean here?

Line 142: Replace the word "univocity" to better reflect what you want to say.

Line 147: "without consent". Did scientists ever talk about doing it without consent?

Line 196: Use "just" instead of "accept"

Line 198: Delete "For" and start the sentence with "Some".

Line 201 "acknowledged (not "es").

Line 206 "to regain a gasp"

Line 223: the sentence ending with "necessary", add references.

Line 234: delete "and" and use ";" instead

Line 242: delete "for long"

Line 243: delete "as it still often is"

Line 244: should instead of shall

Line 265: "rightfulness" - what do you mean by this? Please elaborate.

Line 267 add "in", that is, "helps in making"

Line 268: should be "consensus is built and evolves"

Line 272-273: I don't understand the point here. This sounds like it would only fuel conspiracy.

Line 277: use "to" instead of "with" - "relate to air"

Line 278: English word "confidence"

Line 286: construction instead of fabrication

Line 287: "do so" instead of "go"

Line 291-293: reference needed. Is this McKernon?

Line 296: "Disintermediation" is not the term used in this context, either explain why you borrow it from a different field or replace.

Line 298: "tap in" not "tap on"

Line 302-310: Not novel thoughts.

Communications Earth & Environment is committed to improving transparency in authorship. As part of our efforts in this direction, we are now requesting that all authors identified as 'corresponding author' create and link their Open Researcher and Contributor Identifier (ORCID) with their account on the Manuscript Tracking System prior to acceptance. ORCID helps the scientific community achieve unambiguous attribution of all scholarly contributions. You can create and link your ORCID from the home page of the Manuscript Tracking System by clicking on 'Modify my Springer Nature account' and following the instructions in the link below. Please also inform all co-authors that they can add their ORCIDs to their accounts and that they must do so prior to acceptance.

Version 1:

Decision Letter:

Dear Dr Dorthe,

Your manuscript titled "Engines of connection: How conspiracy theories can enrich the public debate on emerging technologies" has now been seen by our reviewers, whose comments appear below. In light of their advice we are delighted to say that we are happy, in principle, to publish a suitably revised version in Communications Earth & Environment.

We therefore invite you to revise your paper one last time to address the remaining concerns of our reviewers. Specifically, please describe your methodology in sufficient detail for an audience outside the field of ethnography to understand what was done and how. Please also provide as much information about your data base of interviews and transcripts as possible without violating privacy requirements (e.g. reproduction (as Supplementary Information) of the social media pages and blogs that were investigated; number of interviewees etc).

At the same time we ask that you edit your manuscript to comply with our format requirements and to maximise the accessibility and therefore the impact of your work.

EDITORIAL REQUESTS:

****Please take care to match our formatting and policy requirements. We will check revised manuscript and return manuscripts that do not comply. Such requests will lead to delays. ****

SUBMISSION INFORMATION:

In order to accept your paper, we require the files listed at the end of the Editorial Requests Table; the list of required files is also available at <https://www.nature.com/documents/commsj-file-checklist.pdf> .

OPEN ACCESS:

Communications Earth & Environment is a fully open access journal. Articles are made freely accessible on publication. For further information about article processing charges, open access funding, and advice and support from Nature Research, please visit <https://www.nature.com/commsenv/open-access>

Link Redacted

Best regards,

Heike Langenberg, PhD
Chief Editor
Communications Earth & Environment

On X(Twitter): @CommsEarth

REVIEWERS' COMMENTS:

Reviewer #1 (Remarks to the Author):

I appreciate that this revised manuscript brings in method, as requested in the last round of review. I don't know how the editorial staff or the audience for this journal approaches ethnography, but I appreciated it because it offers insight into where the author is coming from and how they make their claims (given that the other reviewer brought up evidence to substantiate the arguments, that seems important).

This paper reads quite differently from the first version, but I thought it was interesting, novel and convincing. The argument that is articulated clearly in lines 348-350 could be reiterated more clearly up front. The new lead, which is much more prosaic, might have the reader missing the argument.

Reviewer #2 (Remarks to the Author):

Thank you for the revised version of the manuscript. There are a few remaining issues:

Methods

I would still like to see more detail about the following:

More information about "patchwork ethnography": I have never heard of this method before and I suspect many other readers may be ignorant as well. Some more information would help.

Some information about data analysis would be helpful. Coding? Maybe the overview of patchwork ethnography will also explain the approach to data analysis.

This sentence was confusing: "Observation of social media pages or blogs, approximately a dozen for each community (solar geoengineering since 2015, COVID vaccines since 2021)." How was data for analysis selected? For example, a random selection of posts?

"(2) Analysis of literature on conspiracy theories and strategies for combating or debunking them" – Any method used to select the literature?

“(3) semi-structured interviews with approximately a dozen individuals, often identified through the snowballing technique, that provide opportunities to explore specific aspects in greater depth.” – Please double-check your notes/transcriptions to locate the exact number of research subjects.

“Diplomacy”

While I appreciate the diplomatic perspective, diplomacy sometimes slips into credulousness. For example, “they circulate and discuss scientific papers, preprints, graphs, and microscope renderings that bear the weight of evidence and are expected to bring credibility to their broader concerns. Talking to conspiracy theorists can be overwhelming because of the amount of scientific evidence that they provide.” The papers they distribute bear the weight of evidence of what? Are they providing scientific evidence, or are they mischaracterizing the information provided in scientific papers etc.? Again, I understand the methodological neutrality but sometimes this seeming impartiality actually presents the arguments of conspiracy theorists as fact.

Solar geoengineering is an umbrella term

“solar geoengineering involves the spraying of particles into the stratosphere, forming a thin layer of aerosols capable of reflecting a portion of the sun’s radiation.”

Technically, this is one kind of solar geoengineering called stratospheric aerosol injection. There are other forms of, and proposals for, solar geoengineering such as putting giant mirrors in space.

Reviewer #3 (Remarks to the Author):

This is an improvement over the previous version. Thank you.

RE: Decision on manuscript COMMSENV-23-0056

I took a little longer than expected, due to the charms of the life of a postdoc, changing jobs, moving countries, teaching duties and other commitments.

I have finally prepared a new manuscript, in the form of a research article rather than a perspective paper, following reviewer 2's suggestion, as endorsed by the editors.

I have taken the very useful reviewers' comments into account as much as possible while essentially rewriting the whole document from scratch, with a new title and the intention of presenting a more compelling argument.

REVIEWER COMMENTS:

Reviewer #1 (Remarks to the Author):

This perspective piece argues that conspiracy theories need to be studied not just as manifestations with concerns about “trust in science”, but that they have to do with norms and values about the power of individual bodies; that they can be seen as “manifestations of disenfranchisement towards scientific promises.”

I am not sure, frankly, how novel the main claim is, as I have not read everything in the conspiracy theory literature within the field of communications / media studies. There are a few different fields which may have treated this topic (sociology, anthropology, comms, STS, and so on). Certainly, within science and technology studies (a field I am more versed in), there are discussions not just about trust in experts, but in power relations. So the more general argument in the paper may have been made.

I essentially rewrote the whole paper with a different angle, and a hopefully more original argument.

However, it is new to study these two examples – solar geoengineering / chemtrails and antivaccines – together. In my view, this makes it enough of a novel contribution to be worth publishing, insomuch as it could help scientists (who seem to be an intended audience) working in these fields understand some of the responses to their work. (I would note that phrases like “entanglement of norms and values with expertise” might be a bit intangible for this audience, and that it would be worth scanning the paper for social science jargon.)

My main, easy-to-address comment is that it would be helpful to have a bit more about the methods / fieldwork up front – even though it's a Perspective, it alludes to empirical material, and that would be useful to contextualize (when, where, etc.).

Added!

A second concern with the manuscript as it is is that it takes some time to describe solar geoengineering and make reference to that, but actually, I believe the chemtrails conspiracy predates discussions of solar geoengineering and arose independently of it. Then, the chemtrails community latched onto solar geoengineering as it emerged. This could be an important point for the argument (and a way that it is different from the mRNA nanobots) – I don't think the chemtrails derived from the technoscientific promises of geoengineering. People thought they were being sprayed for a variety of purposes, mind control, and so on – solar geoengineering entered much later as an additional rationale for the spraying. I am not sure that anyone has written an authoritative history of chemtrails, though, so I can't back this up with a citation – this is simply my understanding from following the topics for many years.

That's interesting and useful to make my argument clearer.

A third comment is that I found myself wishing for a stronger conclusion that could advise the reader about how “a comprehensive dialogue between social and natural sciences, and between scientific communities and the broader public”, that cultivates robustness of discourses, etc., could happen – what are the specific forms or policy actions or cultural changes needed, who does what to develop this, etc.

Added some more specific suggestions on that matter.

In summary, I think this essay could be a useful contribution with some modest changes. The main point – that it's not just about “trusting the science” – is one that needs to be made in a variety of ways for people to truly operationalize it, and so I think it is worth making it for this audience, with regards to these two topics.

Minor points:

- The sentence on p. 9, line 269 seems incomplete or jarring; consider rewording, and I assume the word “Respond” should be “Responses.”

Rephrased.

- p. 10, line 297-8 reminds me of Jill Lepore's book *These Truths*, which is worth reading.

Thanks!

- The title and the concluding sentence with the metaphor of aerosolization just don't work for me; I wish they did and I can't explain why they don't.

I ended up changing the title.

Something of an aside for the authors to consider: The point on p. 6 about the sharp opposition between science and anti-science being misleading, and about conspiracy scientists being interested in science, is important. I am reminded of the observation by Thom Davies and Alice Mah in their book *Toxic Truths: Environmental Justice and Citizen Science in a Post-Truth Age*: “Environmental justice activists typically adopt dual orientations towards science, of mistrust and reliance: (1) challenging the methods, questions, and uses of science, particularly in the context of vested corporate interests, while (2) relying on science itself, as a necessary tool to make investigations, provide evidence, and make arguments” (2020: 10). I suppose the authors might rather separate this population of study – chemtrailers and anti-vaxxers – from EJ activists and citizen scientists, but the ways in which science and “lived experience” are in tension in both areas are kind of interesting, and why some people become or are seen as activists and citizen scientists rather than conspiracy thinkers would be an interesting area of study. There is a lot to say about the intersection of both these with alternative medicine, too, which p. 8 touches on a bit, but there's perhaps more literature there.

This is very useful, thank you! I added this reference in the text.

Reviewer #2 (Remarks to the Author):

Thank you for the opportunity to review, "Aerosolization of trust in science." On the one hand, the manuscript has the potential to make an important contribution to the literature on the social dimensions of conspiratorial thinking, public perceptions of science, and science communication. However, I don't think the "perspective" format is an appropriate model to do this effectively for reasons explained below.

The manuscript does not engage deeply with literature on the social aspects of geoengineering (GE), conspiracies about chemtrails, vaccine-related conspiracy theories, or conspiracy theories in general. This is somewhat understandable given that it is a “perspective” rather than an article. However, the brevity of engagement with existing literature is coupled with a lack of evidence about the central argument of the paper, that:

“Most researchers and university administrators are familiar with that phenomenon, but, while overdetermining the future with grand narratives of power, promises might instill a feeling of dispossession in some parts of the public, who might struggle to make sense of them, or defy them as yet another ideological instrument for maintaining power imbalances (Adam and Groves 2007; Santos 2014). Crafted to generate trust and faith in progress, they turn into threats due to the univocity of this progress: “Elevated levels of expectation and confidence also have the effect of inflaming concerns about risk in different communities based on differing values, knowledges or institutional and organizational forms” (Borup et al. 2006, 292). Conspiracy theories about nanobots in mRNA Covid-19 vaccines, seen in this light, derive from the promises made by scientists and entrepreneurs, imbuing fears that advanced technologies might be implanted in bodies without consent.”

My largest concern with this paper is there isn't enough evidence presented that would convince readers that this is the dynamic behind conspiracies surrounding chemtrails and/or vaccines. What is the basis for arguing that overhyping nanotech or solar geoengineering is the cause of conspiracies about these and related technologies?

I have reframed my argument, as I didn't mean that promises were the cause of conspiracy theories, but rather that how conspiracy theories pick up on promises is something worth paying attention to.

For example, the following argument is made without any evidence, in the form of citations to previous studies or to primary data collection: "They also play a close attention to the vast literature that is enthusiastically promising nanorobots or praising the merits of graphene as a revolutionary material. A sharp opposition between science and anti-science is thus misleading, as a close look at the communities accused of conspiracy theories shows that they nourish a keen curiosity towards scientific announcements and publications, sometimes at the expense of taking them at face value—while they might be preliminary, falsifiable, or sometimes fraudulent."

The softer claim seems more defensible: "In what follows, I explore how differing values or visions of collective progress shape the reception of technoscientific promises, and how conspiracy theories reframe them in terms of health hazards and political disempowerment." Even if this line of argument is more defensible, there still isn't enough evidence provided. For example, there is evidence provided, via references to past research, that perceptions of real power imbalances fuel conspiratorial thinking.

This would be a great perspective piece if it took the form of either a (1) short literature review as illustrations for a new conceptual framework or (2) a short but highly empirical piece for arguments to emerge from data. Instead, it sits in between, failing to establish empirical credibility or conceptual richness. This is not a criticism of the argument itself, which seems novel and important. The issue is that the perspective format is not conducive to making this argument.

Thank you for this encouraging comment. I rewrote the text as an article.

Other comments:

This sentence from the abstract is confusing: "Building on field work in two communities that tend to mutually reinforce each other, solar geoengineering opponents known as chemtrails and anti mRNA Covid vaccines,"

p. 2 – "the same technoscientific promises can be seen as threatening and enslaving and tend to form the bedrock of conspiracy theories." I would change to "some" conspiracy theories. I can think of many conspiracy theories that are not responses to technoscientific promises.

p. 4 "have been promised since the 1980 by prominent scientists and engineers" – 1980s

Corrected, thanks.

Reviewer #3 (Remarks to the Author):

The paper builds on two cases of conspiracy theory – one related to solar geoengineering and another to Covid vaccines – to offer lessons for scientists on how to navigate social discourse and gain public trust. It certainly is an important issue worth exploring and discussing. Overall, the author stresses the importance of a dialogue between social natural sciences and suggest discourse cultivation on all sides.

The paper makes some interesting points, but readers would be better served by reading Cash et. al (2003) or Cash and Belloy (2020). This paper unfortunately does not offer enough of new thoughts/insights.

Cash, David and Clark, William C. and Alcock, Frank and Dickson, Nancy M. and Eckley, Noelle and Jäger, Jill, Saliency, Credibility, Legitimacy and Boundaries: Linking Research, Assessment and Decision Making (November 2002). Available at SSRN: <https://ssrn.com/abstract=372280> or <http://dx.doi.org/10.2139/ssrn.372280>

Cash, D.W., & Belloy, P. (2020). Saliency, Credibility and Legitimacy in a Rapidly Shifting World of Knowledge and Action. Sustainability.

Comments

All taken into account (when relevant in the revised version)—very helpful!

- Line 11: instead of apprehended I would use comprehended/understood.
- Line 33: anti-mask (delete "s")
- Line 41: references?
- Line 46: It reads as if you equate "chemtrails" and "solar geoengineering".
- Line 49: chemical safety (delete s)
- Line 52: comprehend instead of apprehend.
- Line 61: delete: "social sciences (a field named as"
- Line 65: delete "whom", delete "trustful"
- Line 78: what text?
- Line 82: "on science" instead of "in"
- Line 97-98: grammatical error.
- Line 111 to 112: transition needed.
- Line 112: delete "Still"
- Line 123: "shattering the credibility of nanotechnology" I don't this is a result
- Line 131: "the" instead of "this"
- Line 138: "overdetermining the future" – what do you mean here?
- Line 142: Replace the word "univocity" to better reflect what you want to say.
- Line 147: "without consent". Did scientists ever talk about doing it without consent?
- Line 196: Use "just" instead of "accept"
- Line 198: Delete "For" and start the sentence with "Some".
- Line 201" acknowledged (not "es").
- Line 206 "to regain a gasp"
- Line 223: the sentence ending with "necessary", add references.
- Line 234: delete "and" and use ";" instead
- Line 242: delete "for long"
- Line 243: delete "as it still often is"
- Line 244: should instead of shall
- Line 265: "rightfulness" - what do you mean by this? Please elaborate.
- Line 267 add "in", that is, "helps in making"
- Line 268: should be "consensus is built and evolves"
- Line 272-273: I don't understand the point here. This sounds like it would only fuel conspiracy.
- Line 277: use "to" instead of "with" - "relate to air"
- Line 278: English word "confidence"
- Line 286: construction instead of fabrication
- Line 287: "do so" instead of "go"
- Line 291-293: reference needed. Is this McKernon?
- Line 296: "Disintermediation" is not the term used in this context, either explain why you borrow it from a different field or replace.
- Line 298: "tap in" not "tap on"
- Line 302-310: Not novel thoughts.